# Pretreatment Proteinuria Predicts the Prognosis of Patients Receiving Systemic Therapy for Unresectable Hepatocellular Carcinoma

**DOI:** 10.3390/cancers15102853

**Published:** 2023-05-21

**Authors:** Kazuyuki Mizuno, Norihiro Imai, Takafumi Yamamoto, Shinya Yokoyama, Kenta Yamamoto, Takanori Ito, Yoji Ishizu, Takashi Honda, Teiji Kuzuya, Masatoshi Ishigami, Hiroki Kawashima

**Affiliations:** 1Department of Gastroenterology and Hepatology, Nagoya University Graduate School of Medicine, Tsurumai-Cho, Showa-Ku, Nagoya-shi 466-8560, Aichi-ken, Japan; 2Department of Gastroenterology and Hepatology, Fujita Health University, Toyoake 470-1192, Aichi-ken, Japan

**Keywords:** proteinuria, hepatocellular carcinoma, systemic therapy, sorafenib, lenvatinib, atezolizumab plus bevacizumab

## Abstract

**Simple Summary:**

Proteinuria is a common adverse event of systemic therapy for hepatocellular carcinoma (HCC); however, its effect on clinical outcomes is not well understood. A retrospective analysis of 321 patients with unresectable HCC who received systemic therapy as first-line treatment was performed to assess the impact of pretreatment proteinuria on treatment response. Patients without pretreatment proteinuria who received lenvatinib or atezolizumab plus bevacizumab had longer overall survival, but those treated with sorafenib did not. Additionally, a quantitative analysis of 111 patients treated with lenvatinib or atezolizumab plus bevacizumab revealed that the severity of proteinuria was an independent predictor of prognosis, along with liver function. Therefore, pretreatment proteinuria may predict a poorer prognosis in patients with unresectable HCC treated with lenvatinib or atezolizumab plus bevacizumab but not in those treated with sorafenib.

**Abstract:**

Background: Proteinuria is a common adverse event in systemic therapy for hepatocellular carcinoma (HCC). However, whether the presence of pretreatment proteinuria affects the clinical course is still unclear. Method: From 2011 to 2022, 321 patients with unresectable HCC who were treated with systemic therapy as first-line treatment were enrolled in this study. We retrospectively analyzed the presence of pretreatment proteinuria and the treatment course of systemic therapy. Results: In the cohort, 190 patients were tested for proteinuria qualitatively within 3 months before systemic therapy; 75 were treated with sorafenib, 72 were treated with lenvatinib, and 43 were treated with atezolizumab plus bevacizumab. Overall survival tended to be longer for patients treated with lenvatinib and significantly longer with atezolizumab plus bevacizumab in patients without pretreatment proteinuria but not for those treated with sorafenib. Further analysis was performed in 111 patients treated with lenvatinib or atezolizumab plus bevacizumab who had proteinuria measured quantitatively. Multivariate analysis including proteinuria, liver function, and HCC stage revealed that the severity of proteinuria was an independent predictor of prognosis. Conclusion: Pretreatment proteinuria predicts a poorer prognosis in patients with unresectable HCC treated with lenvatinib or atezolizumab plus bevacizumab but not in those treated with sorafenib.

## 1. Introduction

Primary liver cancer was the sixth most commonly diagnosed cancer and the third leading cause of cancer death worldwide in 2020. Primary liver cancer includes hepatocellular carcinoma (HCC) (comprising 75–85% of cases) and intrahepatic cholangiocarcinoma (10–15%) as well as other uncommon types of liver cancer [1]. HCC originates from hepatocytes and usually develops in the presence of chronic liver diseases, such as chronic hepatitis B or C infection, alcohol abuse, and non-alcoholic fatty liver disease [2,3,4,5]. Symptoms of HCC may include fatigue, abdominal pain or swelling, unexplained weight loss, and jaundice. However, most patients with HCC experience no symptoms in the early stages of the disease. Therefore, periodic screening tests, such as ultrasonography, computed tomography (CT), magnetic resonance imaging (MRI), and blood tests are crucial for patients with chronic liver disease. Treatment options for HCC depend on the stage, preserved liver function, and the patient’s performance status [6]. Currently, in early-stage HCC treatment, several options are available, namely ablation, resection, liver transplantation, and transarterial chemoembolization. However, for patients with advanced-stage HCC, treatment options are limited to systemic therapy [7].

Recently, systemic therapy has revolutionized the treatment of unresectable HCC, leading to increased overall survival. This has been demonstrated in two randomized phase 3 trials (the SHARP trial and the Asian Pacific trial), which showed that sorafenib, an oral multikinase inhibitor of the serine-threonine kinases Raf-1 and B-Raf and the receptor tyrosine kinase activity of vascular endothelial growth factor (VEGF) and platelet-derived growth factor, is effective in treating unresectable HCC. The SHARP trial showed that median survival and the time to radiological progression were nearly 3 months longer for patients treated with sorafenib than for those given a placebo in patients with advanced HCC. As a result, sorafenib was approved by the Food and Drug Administration (FDA) for the treatment of HCC in October 2007 [8,9]. During the decade after 2007, no drug showed results indicating that it could be used as a first-line treatment for HCC. Subsequently, the REFLECT trial demonstrated that lenvatinib, an oral multikinase inhibitor that targets the VEGF receptor, fibroblast growth factor receptor, platelet-derived growth factor receptor, RET, and KIT, was non-inferior to sorafenib regarding overall survival and had a longer median progression-free survival than sorafenib in untreated advanced HCC. On the basis of this result, the FDA approved lenvatinib for the treatment of HCC in August 2018 [10]. Subsequently, the IMbrave150 trial demonstrated a statistically significant improvement in overall survival and progression-free survival with programmed death ligand 1 inhibitor, atezolizumab, and the monoclonal antibody bevacizumab, which targets VEGF, compared with sorafenib. In May 2020, the FDA approved atezolizumab in combination with bevacizumab for patients with unresectable or metastatic HCC [11,12]. Recently, the HIMALAYA trial showed that tremelimumab plus durvalumab significantly improved overall survival compared with sorafenib and that durvalumab monotherapy was non-inferior to sorafenib. On the basis of the results of this trial, the FDA approved tremelimumab plus durvalumab for patients with unresectable HCC in October 2022 [13]. Currently, the Barcelona Clinic Liver Cancer (BCLC) and the European Association for the Study of the Liver position paper recommend atezolizumab plus bevacizumab, and when there are contraindications to this combination, sorafenib or lenvatinib is recommended as the first-line treatment for patients with HCC [7,14].

In addition to the immune-related adverse events noted with immune checkpoint inhibitors [15,16], bevacizumab-induced proteinuria is a common problem in patients who receive atezolizumab plus bevacizumab. Bevacizumab is a recombinant humanized monoclonal antibody that targets VEGF activity, inhibiting binding to its receptors, VEGF receptor 1 and VEGF receptor 2 [17]. Proteinuria with bevacizumab has been previously reported in various studies (21–41%), and IMbrave150 reported an incidence of 20% for all grades of proteinuria and 3% for grade ≥ 3 with treatment with atezolizumab plus bevacizumab [11,12,18,19,20]. The exact cause of bevacizumab-induced proteinuria is not well known, but it is thought to be related to nitric oxide inhibition, increased peripheral vascular resistance, renal dysfunction, and glomerular damage caused by the inhibition of VEGF produced by podocytes [21]. Proteinuria is also a problematic adverse event with sorafenib and lenvatinib therapy. In the REFLECT trial, any grade of proteinuria and grade ≥ 3 proteinuria were observed in 11% and 2% of patients, respectively, in the sorafenib arm, with rates of 25% and 6%, respectively, in the lenvatinib arm [10]. In a phase II trial of Asian patients with HCC, proteinuria was the most frequently reported adverse event leading to lenvatinib discontinuation [22]. Thus, proteinuria is a serious adverse event associated with current systemic therapy for HCC that must be strictly controlled. However, it is still unclear whether the presence of pretreatment proteinuria affects the clinical course. Therefore, we hypothesized that the presence of proteinuria before systemic therapy may affect the prognosis in patients with unresectable HCC.

## 2. Patients and Methods

### 2.1. Study Population and Data Collection

Between January 2011 and October 2022, 321 consecutive patients who received sorafenib, lenvatinib, or atezolizumab plus bevacizumab for unresectable HCC as first-line systemic therapy at Nagoya University Hospital were included in this study. Among them, 190 patients were selected using the following inclusion criteria: (1) qualitative urine protein measurement at least 3 months before drug administration and (2) treatment continued for at least 1 week if no adverse events occurred.

Baseline characteristics, namely age, sex, underlying liver disease, laboratory data, and tumor-specific characteristics, such as drug administration period, tumor stage at administration in accordance with the BCLC classification, macrovascular and portal vein invasion, and extrahepatic spread, were assessed retrospectively. 

The severity of proteinuria was graded in accordance with the National Cancer Institute Common Terminology Criteria for Adverse Events version 5.0 [23].

This study received approval from the institutional review board of our hospital (protocol number: 2021-0247) and adhered to ethical standards established by relevant committees on human experimentation (institutional and national) and the Helsinki Declaration of 1975.

### 2.2. Underlying Liver Diseases

The etiologies of HCC were classified as follows: positive for anti-hepatitis C virus (HCV) antibody: HCV; positive for hepatitis B virus (HBV) surface antigen: HBV; and negative for both anti-HCV antibody and HBV surface antigen: NBNC.

### 2.3. Liver Function Assessment

We used the Child–Pugh classification [24] and albumin–bilirubin (ALBI) score to assess liver function. Briefly, the ALBI score was calculated on the basis of laboratory data using the following formula:ALBI score = log10 bilirubin (µmol/L) × 0.66 + albumin (g/L) × −0.085.

Patients were assigned to one of three groups on the basis of their ALBI score as follows: ALBI grade 1 (ALBI score ≤ −2.60), grade 2 (<−2.60 to  ≤ −1.39), and grade 3 ( > −1.39). Lower grades correspond to better liver function [25]. To gain a more thorough understanding of patients with an ALBI grade of 2, we used a revised grading system that included four levels, with sub-grading for the middle grade of 2 (2a and 2b), with an ALBI score of −2.27 as the cutoff. This sub-grading system was previously developed on the basis of a value for indocyanine green retention after 15 min of 30% [26].

### 2.4. HCC Diagnosis 

HCCs were mainly diagnosed by hemodynamic imaging, such as contrast-enhanced CT, gadolinium ethoxybenzyl diethylenetriaminepentaacetic acid (Gd-EOB-DTPA)-enhanced MRI, and/or contrast-enhanced ultrasonography with perflubutane (Sonazoid^®^; Daiichi Sankyo Co., Ltd., Tokyo, Japan). HCC was diagnosed pathologically only for inconclusive cases.

### 2.5. Systemic Therapy

Patients received systemic therapy until confirmed disease progression, unacceptable adverse events, consent withdrawal, or a physician’s decision based on a patient’s condition and clinical data. Generally, sorafenib was used until February 2018, lenvatinib was used from March 2018 to October 2020, and atezolizumab plus bevacizumab therapy was used thereafter; however, the choice of agents was also determined on the basis of complications. The systemic therapy regimens were as follows: 400 mg sorafenib orally twice daily; 12 mg lenvatinib for a body weight of ≥60 kg or 8 mg for a body weight of <60 kg orally daily; and 1200 mg atezolizumab plus 15 mg/kg bevacizumab intravenously every 3 weeks.

In principle, treatment effects were examined every 2–4 months using contrast-enhanced CT or Gd-EOB-DTPA-enhanced MRI using the RECIST and mRECIST criteria [27,28,29]. We also assessed tumor markers, such as α-fetoprotein and des-γ-carboxy prothrombin, at every patient visit. We used the Common Terminology Criteria for Adverse Events version 5.0 to evaluate the adverse events. Mainly, we interrupted or discontinued a drug in accordance with the guidelines for treatment provided by the manufacturer. For proteinuria, the patient was informed of the risks of continued therapy with bevacizumab, which was continued with a urine protein/creatinine ratio (UPCR) of ≤3.5, if necessary.

### 2.6. Statistical Analysis

Continuous variables, expressed as medians (interquartile ranges), were compared using the Mann–Whitney U test, while categorical variables, expressed as numbers (percentages), were compared using the Chi-squared test. Overall survival was calculated from the start date of first-line systemic therapy to death or the date of the last follow-up. 

Survival outcomes were assessed using the Kaplan–Meier method, and the log-rank test was used to compare the differences between subgroups. Statistical significance was defined at *p*  <  0.05. A Cox proportional hazards model was used for univariate and multivariate analyses of factors related to overall survival. The analyzed variables were patient age and sex, etiology (HBV/HCV/NBNC), ALBI score, Child–Pugh score, BCLC stage (A/B/C), UPCR, and first-line treatment. Because both the ALBI score and Child–Pugh score were variables for liver function, we used only the ALBI score in the multivariate analyses.

The cumulative incidence rate of grade ≥ 3 proteinuria in 1 year was calculated from the initiation date of the first-line systemic therapy to the incidence of proteinuria or the date of the last follow-up or end of any systemic therapy in 1 year. The cumulative incidence rate was estimated using the Kaplan–Meier method, with stratification into two groups by the presence or absence of pretreatment proteinuria.

All statistical analyses were performed using EZR (Saitama Medical Center, Jichi Medical University, Saitama, Japan), which is a graphical user interface for R (www.r-project.org). Specifically, EZR is a modified version of R commander that includes statistical functions commonly used in biostatistics [30].

## 3. Results

### 3.1. Patients Characteristics and Proteinuria

Table 1 shows the patients’ characteristics. The patients’ median age was 72 years, and 156 (82.1%) were male. Regarding HCC stage, 4 (2.1%) were BCLC A, 67 (35.3%) were BCLC B, and 119 (62.6%) were BCLC C. Regarding liver function, 38 (20.0%) were modified (m)ALBI grade 1, 61 (32.1%) were mALBI grade 2a, 86 (45.3%) were mALBI grade 2b, and 5 (2.6%) were mALBI grade 3. Seventy-five patients (39.5%) received sorafenib as the first-line treatment, 72 patients (37.9%) received lenvatinib, and 43 patients (22.6%) received atezolizumab plus bevacizumab. 

To test our hypothesis that pretreatment proteinuria may affect the treatment course of HCC, we divided the patients into two groups on the basis of urine dipstick test results as follows: pretreatment qualitative proteinuria-negative and “+or greater”. Serum creatinine was significantly higher (*p* = 0.004), NBNC was significantly more frequent (*p* = 0.030), and mALBI grade was significantly worse (*p* = 0.049) in the pretreatment proteinuria-positive group compared with the pretreatment proteinuria-negative group.

When comparing the backgrounds, HCC stage, and first-line treatment in patients with pretreatment proteinuria, there were no significant differences compared with those without proteinuria. However, liver and renal functions were slightly worse in those with proteinuria compared with those without proteinuria, probably owing to differences in comorbidities.

### 3.2. Overall Survival

The median overall survival was 13.6 months (95% confidence interval (CI): 7.8–16.8) for the 75 patients treated with sorafenib, 16.7 months (95% CI: 12.3–25.9) for the 72 patients treated with lenvatinib, and 21.9 months (95% CI: 12.5–not available) for the 43 patients treated with atezolizumab plus bevacizumab. 

Next, we investigated the impact of the presence of pretreatment proteinuria on overall survival for each first-line treatment (Figure 1). Surprisingly, although there was no difference in overall survival between patients with and without pretreatment proteinuria in patients treated with sorafenib, overall survival tended to be longer for patients treated with lenvatinib and significantly longer for those treated with atezolizumab plus bevacizumab in patients without pretreatment proteinuria (lenvatinib: *p* = 0.075, atezolizumab plus bevacizumab: *p* = 0.009). 

To obtain further insights regarding the clinical impact of pretreatment proteinuria, we evaluated the UPCR. Recent studies have reported that evaluating the UPCR might be appropriate for assessing proteinuria in patients with HCC or thyroid cancer who receive tyrosine kinase inhibitors, including lenvatinib [31,32,33]. In total, 121 patients in this study underwent urine dipstick tests and UPCR measurement, especially after 2018, when lenvatinib was approved. The results showed that in most of the patients, the results of the two tests were correlated; however, there were some outliers (Figure 2A). Therefore, as previously reported [34], we defined proteinuria as UPCR ≥ 0.15 g/g creatinine.

On the basis of proteinuria defined by UPCR, overall survival was reanalyzed (Figure 2B–D). Consistent with our findings using qualitative proteinuria measurements, overall survival was significantly shorter in patients with proteinuria defined by UPCR compared with that in patients without proteinuria with first-line treatment with both lenvatinib (Figure 2C; *p* = 0.004) and atezolizumab plus bevacizumab (Figure 2D; *p* = 0.039).

These results suggested that pretreatment proteinuria may predict a poorer prognosis in patients with unresectable HCC treated with lenvatinib or atezolizumab plus bevacizumab compared with sorafenib.

### 3.3. Multivariate Analysis of Overall Survival

Multivariate analysis was performed to determine whether proteinuria is an independent prognostic factor in systemic therapy for HCC. Because our results showed distinct clinical outcomes associated with proteinuria between patients treated with sorafenib and the other drugs, we performed a multivariate analysis in patients who received first-line treatment with lenvatinib or atezolizumab plus bevacizumab. In the selected cohort, the reanalyzed median overall survival was 21.9 months (95% CI: 15.2–25.9). Regarding the patients’ characteristics, univariate analysis identified ALBI score (per 1 index increase: hazard ratio (HR): 4.01; *p* < 0.001), Child–Pugh score 7–9 (vs. 5–7: HR: 3.50, *p* = 0.002; vs. 8: HR: 13.96, *p* < 0.001; vs. 9: HR: 52.44, *p* < 0.001), and UPCR (per 1 g/g creatinine increase: HR: 1.71; *p* = 0.002) as potential prognostic factors for poor overall survival (Table 2). Next, we used multivariate Cox proportional hazards models and identified ALBI score (per 1 index increase: HR: 4.96; *p* < 0.001) and UPCR (per 1 g/g creatinine: HR: 1.70; *p* = 0.006) as significant independent prognostic factors (Table 2). These results suggested that pretreatment proteinuria was an independent predictor of prognosis in patients treated with lenvatinib and atezolizumab plus bevacizumab.

### 3.4. Incidence of Proteinuria Based on Pretreatment Proteinuria

Because our data suggested that pretreatment proteinuria is an independent predictor for systemic treatment for HCC, we evaluated whether the presence of proteinuria worsened during the treatment course. The Kaplan–Meier curves for the cumulative incidences of grade ≥ 3 proteinuria 1 year after starting lenvatinib or atezolizumab plus bevacizumab are shown in Figure 3. The cumulative incidence of grade ≥ 3 proteinuria was significantly higher in patients with pretreatment proteinuria compared with those without proteinuria in patients with first-line treatment with lenvatinib and with atezolizumab plus bevacizumab (lenvatinib: *p* = 0.0444; atezolizumab plus bevacizumab: *p* = 0.0107).

It is not surprising that the presence of pretreatment proteinuria predicted the incidence of grade ≥ 3 proteinuria in systemic therapy for HCC; however, our results emphasize the importance of proteinuria evaluation before systemic therapy and careful observation thereafter.

## 4. Discussion

Proteinuria is a common adverse event in molecular targeted agent therapy for HCC. However, whether the presence of pretreatment proteinuria affects the clinical course is unclear. In this study, we evaluated 190 patients with unresectable HCC who were treated with first-line treatment with sorafenib, lenvatinib, or atezolizumab plus bevacizumab. We revealed that the presence of pretreatment proteinuria was a significant predictor for a poor prognosis in patients treated with lenvatinib or atezolizumab plus bevacizumab, but it was not a significant predictor in patients treated with sorafenib. Multivariate analysis of 111 patients treated with lenvatinib or atezolizumab plus bevacizumab who had quantitative proteinuria measurements revealed that the severity of pretreatment proteinuria was an independent predictor of prognosis. For these reasons, we propose that pretreatment proteinuria predicts a poor prognosis in patients with unresectable HCC treated with lenvatinib or atezolizumab plus bevacizumab compared with sorafenib. 

In this study, we identified that pretreatment proteinuria was a prognostic factor independent of liver cancer progression and liver function in patients who were treated with lenvatinib or atezolizumab plus bevacizumab (HR: 1.70; 95% CI: 1.16–2.48; *p* = 0.006). Both lenvatinib and bevacizumab have anti-VEGF activity.

In the kidneys, the filtration of plasma within nephrons occurs at the glomerular filtration barrier located in the glomerular capillary beds. This barrier comprises three layers: fenestrated endothelial cells, the basement membrane, and the foot processes of the visceral epithelial cells (podocytes). VEGF factor A (VEGFA) is expressed by both podocytes [35] and renal tubular epithelial cells [36]. VEGFA binds to VEGF receptor 1 and VEGF receptor 2 [37], which are mainly located in the glomerular and peritubular capillary endothelium [36]. Previous studies have suggested that the maintenance of glomerular endothelial integrity is heavily dependent on the precise regulation of paracrine VEGFA–VEGFR2 signaling between the podocyte and renal endothelium and that anti-VEGF therapy can lead to renal endothelial injury, primarily manifested as proteinuria, hypertension, and renal-specific thrombotic microangiopathy [38,39]. The reported incidence of proteinuria after bevacizumab therapy ranges from 21% to 62%, with the greatest risk associated with high-dose therapy [18]. Additionally, bevacizumab use was associated with the development of renal thrombotic microangiopathy with biopsy findings of glomerular endothelial cell injury [18].

Similarly, the reported incidences of all-grade and grade ≥ 3 proteinuria with solid tumors treated with VEGFR tyrosine kinase inhibitors, such as lenvatinib, were 18.7% and 2.4%, respectively [40]. Interestingly, in contrast to renal biopsy specimens from patients treated with bevacizumab, biopsy specimens for the majority of the patients receiving tyrosine kinase inhibitors exhibited podocytopathies, including minimal change disease and collapsing focal glomerular sclerosis [41].

HCC is common in cirrhotic livers. In patients with liver cirrhosis, scar tissue compresses blood vessels inside the liver and leads to increased portal vein pressure and decreased albumin synthesis owing to impaired liver function, which results in fluid accumulation, such as ascites [42,43]. Proteinuria also causes hypoalbuminemia through the loss of protein into the urine, which can lead to edema [44]. Therefore, adverse events associated with proteinuria in patients with cirrhosis may exacerbate fluid retention due to hypoalbuminemia.

In this study, patients without pretreatment proteinuria had a higher likelihood of receiving second-line systemic therapy after completing first-line systemic therapy compared with those with pretreatment proteinuria. Specifically, 54.3% (31/57) of the patients without pretreatment proteinuria received second-line systemic therapy, whereas only 31.0% (9/29) of patients with pretreatment proteinuria received this therapy. However, this difference did not reach statistical significance (*p* = 0.068). 

Furthermore, this study found that patients without proteinuria who received lenvatinib as first-line therapy tended to have higher relative dose intensities (RDIs) at 4 weeks and 8 weeks compared with those with proteinuria (4-week RDI: *p* = 0.054, 8-week RDI: *p* = 0.093) (Appendix A). Additionally, patients without pretreatment proteinuria tended to have a longer duration of treatment with first-line atezolizumab plus bevacizumab, without the need to withdraw bevacizumab or terminate atezolizumab plus bevacizumab owing to treatment response or adverse events, compared with patients with pretreatment proteinuria (*p* = 0.053) (Appendix A). 

Proteinuria is a known adverse event associated with anti-VEGF activity and is often linked to a deterioration in quality of life and drug discontinuation. Preserving a high RDI during systemic therapy is crucial for managing HCC, as demonstrated in the findings of this study [45,46].

Notably, unlike with the other two regimens, our results showed no difference in prognosis in patients with or without pretreatment proteinuria treated with sorafenib. This finding might be explained by the lower incidence of proteinuria in patients treated with sorafenib compared with the other two regimens. In the REFLECT trial, which directly compared patients treated with lenvatinib and sorafenib, the overall incidences of any grade and grade ≥ 3 proteinuria associated with lenvatinib and sorafenib were 24.6% and 11.3%, and 5.7%, and 1.7%, respectively [10]. In the IMbrave150 trial, which directly compared patients treated with atezolizumab plus bevacizumab and sorafenib, the overall incidences of any grade and grade ≥ 3 proteinuria associated with atezolizumab plus bevacizumab and sorafenib were 28.9% and 5.1%, and 4.0% and 0.6%, respectively [12]. The lower incidence of proteinuria in patients treated with sorafenib compared with the other therapies may indicate a lower impact on overall survival.

Another reason for the lack of a difference in prognosis with sorafenib with and without proteinuria might be the difference in treatment availability. When sorafenib was used, it was the only systemic therapy regimen available. By contrast, when lenvatinib and atezolizumab plus bevacizumab were used, ramucirumab, regorafenib, and cabozantinib were available as second-line and later regimens. These regimens inhibit VEGF activity. The presence or absence of proteinuria may have also affected the availability of these drugs in patients who received first-line treatment with bevacizumab or atezolizumab plus bevacizumab.

Tremelimumab plus durvalumab was approved by the FDA in October 2022 for adult patients with unresectable HCC. In the HIMALAYA trial, tremelimumab plus durvalumab significantly improved overall survival versus sorafenib. This regimen does not include drugs that inhibit VEGF activity, and in the trial, treatment-related proteinuria was not described [13]. These results suggest that the presence of proteinuria may not reduce the effectiveness of this regimen. Although the presence or absence of proteinuria may help in the choice of first-line treatments, the importance of proteinuria evaluation remains in second-line and subsequent treatments because these regimens have anti-VEGF activity.

There are three limitations in our study. First, this was a single-center, small, retrospective study. Second, few patients with first-line treatment with sorafenib were evaluated by UPCR; therefore, we analyzed the results of the urine dipstick test only. Third, because we examined the data of the first-line regimen, we did not evaluate second-line and further regimens. Therefore, there is a concern that either regimen might have influenced the overall survival of patients who received first-line treatment with these regimens. However, similar results were obtained with both regimens, similar to findings in real-world clinical practice; therefore, we believe that the findings in this study are important.

## 5. Conclusions

In conclusion, pretreatment proteinuria in patients who received lenvatinib or atezolizumab plus bevacizumab as first-line treatment for HCC was a poor prognostic factor independent of liver function and HCC progression. It may be desirable to consider the presence or absence of pretreatment proteinuria in the treatment choice for unresectable HCC. Furthermore, drug-specific comparative studies in patients with proteinuria are warranted.

## Figures and Tables

**Figure 1 cancers-15-02853-f001:**
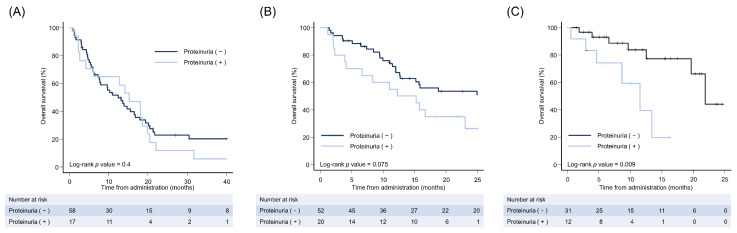
Kaplan–Meier analysis of overall survival based on proteinuria by the urine dipstick test. Kaplan–Meier estimate of overall survival for patients based on proteinuria by urine dipstick test. We defined proteinuria as “+or greater” according to urine dipstick test results. As a first-line treatment, patients were treated with (**A**) sorafenib, (**B**) lenvatinib, or (**C**) atezolizumab plus bevacizumab.

**Figure 2 cancers-15-02853-f002:**
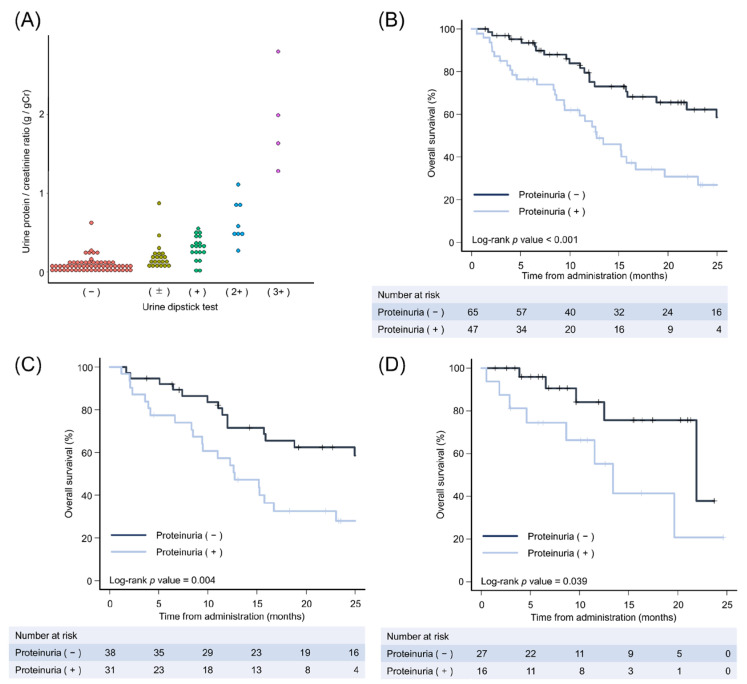
Kaplan–Meier analysis of overall survival on the basis of proteinuria by the urine protein/creatinine ratio. (**A**) Dot plot showing the relationship between the urine dipstick test and the urine protein/creatinine ratio for patients with simultaneous measurements at administration of systemic therapy. (**B**–**D**) Kaplan–Meier estimates of overall survival for patients with proteinuria measured by the urine protein/creatinine ratio. We defined proteinuria as a urine protein/creatinine ratio ≥ 0.15 g/g creatinine. As first-line treatment, patients were treated with (**B**) both lenvatinib and atezolizumab plus bevacizumab, (**C**) lenvatinib, and (**D**) atezolizumab plus bevacizumab.

**Figure 3 cancers-15-02853-f003:**
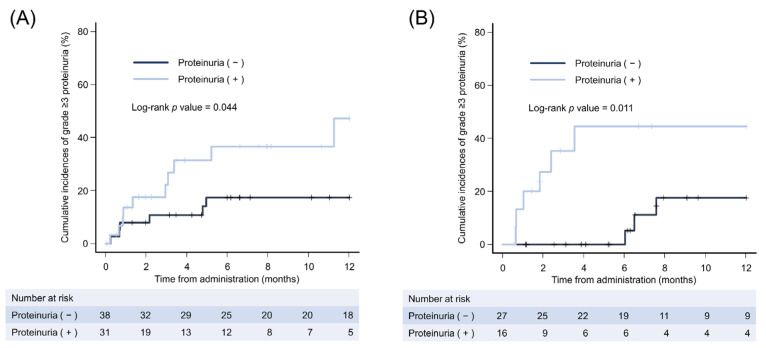
Incidence of proteinuria based on pretreatment proteinuria. Kaplan–Meier curves for the cumulative incidences of grade ≥ 3 proteinuria in 1 year with the administration of (**A**) lenvatinib and (**B**) atezolizumab plus bevacizumab.

**Table 1 cancers-15-02853-t001:** Baseline characteristics of the study patients based on pretreatment proteinuria.

		Total (n = 190)	Pretreatment Proteinuria		*p* Value
			Negative (n = 141)	Positive (n = 49)	
Age (years)		72 [66, 78]	72 [66, 78]	72 [66, 77]	0.800
Sex					
	Female	34 (17.9)	29 (20.7)	5 (10.0)	0.131
	Male	156 (82.1)	111 (79.9)	45 (90.0)	
Barcelona Clinic Liver Cancer stage					
	A	4 (2.1)	3 (2.1)	1 (2.0)	0.782
	B	67 (35.3)	52 (36.9)	15 (30.6)	
	C	119 (62.6)	86 (61.0)	33 (67.3)	
Child-Pugh classification					
	A5	84 (44.2)	66 (46.8)	18 (36.7)	0.123
	A6	78 (41.1)	54 (38.3)	24 (49.0)	
	B7	20 (10.5)	14 (9.9)	6 (12.2)	
	B8	7 (3.7)	7 (5.0)	0 (0.0)	
	B9	1 (0.5)	0 (0.0)	1 (2.0)	
Etiology of chronic liver disease					
	Hepatitis B	28 (14.7)	25 (17.7)	3 (6.1)	0.030
	Hepatitis C	55 (28.9)	44 (31.2)	11 (22.4)	
	Nonviral	107 (56.3)	72 (51.1)	35 (71.4)	
Modified ALBI grade					
	1	38 (20.0)	34 (24.1)	4 (8.2)	0.049
	2a	61 (32.1)	40 (28.4)	21 (42.9)	
	2b	86 (45.3)	63 (44.7)	23 (46.9)	
	3	5 (2.6)	4 (2.8)	1 (2.0)	
First line treatment					
	Atezolizumab plus bevicizumab	43 (22.6)	31 (22.0)	12 (24.5)	0.755
	Lenvatinib	72 (37.9)	52 (36.9)	20 (40.8)	
	Sorafenib	75 (39.5)	58 (41.1)	17 (34.7)	
AFP (ng/mL)		72.5 [6.0, 978.5]	68.5 [6.0, 921.5]	74.0 [5.0, 1159.0]	0.754
ALB (g/dL)		3.6 [3.2, 3.9]	3.6 [3.2, 4.0]	3.5 [3.1, 3.8]	0.073
ALT (U/L)		30 [20, 43]	30 [21, 45]	28 [19, 38]	0.374
AST (U/L)		40 [30, 64]	40 [31, 64]	38 [30, 55]	0.802
CRE (mg/dL)		0.80 [0.66, 1.02]	0.76 [0.64, 0.96]	0.94 [0.72, 1.10]	0.004
PT (%)		89.9 [78.9, 96.2]	88.5 [76.0, 96.1]	91.5 [83.7, 97.1]	0.311
TBIL (mg/dL)		0.9 [0.7, 1.1]	0.9 [0.7, 1.1]	0.7 [0.60, 1.0]	0.023

Continuous variables, expressed as median [interquartile range], were compared using the Mann–Whitney U test. Categorical variables, expressed as number (percentage), were compared using the Chi-squared test. AFP α-fetoprotein, ALB albumin, ALBI albumin-bilirubin, ALT alanine aminotransferase, AST aspartate aminotransferase, Cre creatinine, PT prothrombin time, TBIL total bilirubin.

**Table 2 cancers-15-02853-t002:** Univariate and multivariate analysis of overall survival.

	Univariate Analysis		Multivariate Analysis	
Factor	Hazard Ratio (95%CI)	*p* Value	Hazard Ratio (95%CI)	*p* Value
Age (per year)	1.012 (0.985-1.039)	0.388	1.013 (0.981-1.046)	0.43
Male sex	1.673 (0.839–3.339)	0.144	1.586 (0.779–3.230)	0.204
Etiology of chronic liver disease				
Hepatitis B	Reference		Reference	
Hepatitis C	1.123 (0.467–2.698)	0.795	0.577 (0.225–1.482)	0.253
Nonviral	1.267 (0.556–2.887)	0.573	0.568 (0.226–1.430)	0.23
ALBI score (per 1 increase)	4.005 (0.556–2.887)	<0.001	4.956 (2.300–10.69)	<0.001
Child-Pugh classification				
A5	Reference			
A6	1.670 (0.909–3.070)	0.098		
B7	3.503 (1.614–7.605)	0.002		
B8	13.96 (4.415–44.13)	<0.001		
B9	52.44 (5.585–492.3)	<0.001		
Barcelona Clinic Liver Cancer stage				
A	Reference		Reference	
B	1.839 (0.247–13.68)	0.552	2.089 (0.265–16.49)	0.485
C	2.534 (0.345–18.64)	0.361	2.686 (0.350–20.63)	0.342
Ratio of urinary protein to creatinine	1.709 (1.226–2.383)	0.002	1.697 (1.162–2.481)	0.006
(per 1 g/gCr increase)				
First-line regimen				
Aterolizumab+bevacizumab	Reference		Reference	
Lenvatinib	1.110 (0.582–2.119)	0.751	1.026 (0.523–2.013)	0.94

A Cox proportional hazards model was utilized to conduct univariate and multivariate analyses of factors associated with overall survival using the forced entry method. In order to account for potential confounding factors, we included age, sex, etiology, the ALBI score, HCC stage, and treatment regimen, as these factors are already known to be prognostic factors for patients with HCC. ALBI albumin-bilirubin, CI confidence interval, HCC hepatocellular carcinoma.

## Data Availability

All data generated or analyzed during this study are included in this article. Further inquiries can be directed to the corresponding author.

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
