# Peer review of "Pretreatment Proteinuria Predicts the Prognosis of Patients Receiving Systemic Therapy for Unresectable Hepatocellular Carcinoma"

_cancers, 2023, doi:10.3390/cancers15102853_

Round 1
Reviewer 1 Report
In the retrospective single-center study presented here, the authors were able to show that pretreatment proteinuria is a prognostic factor for outcome after systemic combination treatment with atezolizumab + bevacizumab that is independent of liver function. In the discussion itself, they address the limitations of the study, such as that few patients in the sorafinib group were assessed with the UPCR score. Pre-existing proteinuria cannot yet be used to rethink treatment decisions for non-resectable HCC; further clinical trials are needed for this purpose. The results presented here are clearly structured and also their limitations were discussed, therefore I recommend the manuscript for publication, only a minor revision of the English language is needed.
Author Response
#Reviewer 1
In the retrospective single-center study presented here, the authors were able to show that pretreatment proteinuria is a prognostic factor for outcome after systemic combination treatment with atezolizumab + bevacizumab that is independent of liver function. In the discussion itself, they address the limitations of the study, such as that few patients in the sorafinib group were assessed with the UPCR score. Pre-existing proteinuria cannot yet be used to rethink treatment decisions for non-resectable HCC; further clinical trials are needed for this purpose.
The results presented here are clearly structured and also their limitations were discussed, therefore I recommend the manuscript for publication, only a minor revision of the English language is needed.
Thank you very much for taking the time to review our manuscript. We appreciate your positive feedback and your recognition of the clear structure of our results and the discussion of the limitations of our study. We have carefully reviewed the manuscript to address the minor language revisions you suggested. Thank you again for your valuable feedback and your recommendation for publication.

Reviewer 2 Report
Atezolizumab plus bevacizumab and lenvatinib are drugs used to treat unresectable hepatocellular carcinoma (HCC). Each drug has a different mechanism of action, and it has been suggested that combining several drugs may improve the therapeutic effect of HCC.
However, these drugs also have side effects. In some patients, the drugs may affect the kidneys, causing an increase in urinary protein. In such cases, progressive proteinuria may lead to kidney dysfunction.
Patients undergoing treatment should undergo regular kidney function tests to manage proteinuria. If proteinuria progresses, treatment should be reviewed or considered for discontinuation. In treatment, it is important to continuously monitor the patient's health status and to detect and address side effects as early as possible.
In this study, we retrospectively examined the impact of proteinuria before primary therapy on the efficacy of treatment with atezolizumab plus bevacizumab and lenvatinib. The results showed that patients presenting with proteinuria from the beginning had a significantly poorer prognosis than those who did not stop, making it an independent poor prognostic factor. Although the study has many limitations, such as the medium-sized number of patients used in the study, the particularly small number of cases treated with sorafenib, and the retrospective nature of the study, the study methodology is excellent and the results are extremely useful.As a reviewer, I would like to express my deep appreciation for the opportunity to review the paper at the beginning. I would also like to make the following comments to make the paper even more attractive to readers.
1. Authors should specify, for example, how treatment for HCC was interrupted by the occurrence or worsening of proteinuria. For example, the number of treatment courses, cumlative drug doses, and relative dose intensity must be applicable.
2. For the multivariate analysis in Table 2, the method of putting all univariate factors into the multivariate analysis is not acceptable. You should have an expert comment on it and re-do it.
3. The discussion of proteinuria and the effect of primary HCC treatment is inadequate. The possibility that renal dysfunction as a component of lifestyle-related diseases affected survival should be eliminated. Please review the discussion from this perspective.
Author Response
#Reviewer 2
Atezolizumab plus bevacizumab and lenvatinib are drugs used to treat unresectable hepatocellular carcinoma (HCC). Each drug has a different mechanism of action, and it has been suggested that combining several drugs may improve the therapeutic effect of HCC.
However, these drugs also have side effects. In some patients, the drugs may affect the kidneys, causing an increase in urinary protein. In such cases, progressive proteinuria may lead to kidney dysfunction.
Patients undergoing treatment should undergo regular kidney function tests to manage proteinuria. If proteinuria progresses, treatment should be reviewed or considered for discontinuation. In treatment, it is important to continuously monitor the patient's health status and to detect and address side effects as early as possible.
In this study, we retrospectively examined the impact of proteinuria before primary therapy on the efficacy of treatment with atezolizumab plus bevacizumab and lenvatinib. The results showed that patients presenting with proteinuria from the beginning had a significantly poorer prognosis than those who did not stop, making it an independent poor prognostic factor. Although the study has many limitations, such as the medium-sized number of patients used in the study, the particularly small number of cases treated with sorafenib, and the retrospective nature of the study, the study methodology is excellent and the results are extremely useful.
As a reviewer, I would like to express my deep appreciation for the opportunity to review the paper at the beginning. I would also like to make the following comments to make the paper even more attractive to readers.
Thank you very much for your thoughtful and constructive review of our paper. We greatly appreciate your positive feedback on the study methodology and the usefulness of our results. We also acknowledge the limitations you have pointed out, including the small number of cases treated with sorafenib and the retrospective nature of the study. We thank you for your valuable input and for your appreciation of our work. We have revised the manuscript to make it even more informative and useful to readers.
- Authors should specify, for example, how treatment for HCC was interrupted by the occurrence or worsening of proteinuria. For example, the number of treatment courses, cumulative drug doses, and relative dose intensity must be applicable.
We appreciate your suggestion to provide more detailed information on how treatment for HCC was interrupted by the occurrence or worsening of proteinuria. We have added a section to the manuscript detailing the effect of pretreatment proteinuria on the transition rate to second-line treatment, relative dose intensity, and treatment duration without discontinuation of bevacizumab. (Page 10 line 333) We hope that this addition addresses your concern, and we would be happy to provide any additional information if needed.
- For the multivariate analysis in Table 2, the method of putting all univariate factors into the multivariate analysis is not acceptable. You should have an expert comment on it and re-do it.
Thank you for your suggestion regarding the use of multivariate analysis. We are aware of the various methods, including stepwise, hierarchical, and forced entry.
The purpose of Table 2 was to perform a multivariate analysis to investigate the causal effect of pretreatment proteinuria on the prognosis of patients with HCC who were treated with systemic therapy. To adjust for potential confounding factors, we included age, sex, etiology, the ALBI score, HCC stage, and treatment regimen, because all are already known prognostic factors for patients with HCC. In total, 57 events were recorded in this study, and we entered 7 independent variables, which we believe provided an adequate sample size for analysis.
We acknowledge that stepwise methods are commonly used in multivariate analysis. However, we are also aware that this approach can lead to a large number of calculated P-values before reaching the final model, and reproducibility can be problematic due to excessive examination of the data. To address these concerns, we conducted our multivariate analysis using the forced entry method. We chose this approach because we believe that independent variables should be selected based on prior research and their potential relationship with the dependent variable.
Thank you for your thoughtful review of our work. We hope that these clarifications improve the clarity and scientific rigor of our manuscript.
- The discussion of proteinuria and the effect of primary HCC treatment is inadequate. The possibility that renal dysfunction as a component of lifestyle-related diseases affected survival should be eliminated. Please review the discussion from this perspective.
Thank you for bringing this to our attention. We have added the necessary information to the manuscript. (Page 10 line 326) Thank you once again for your comments and suggestions, which have helped to improve the quality of our work. We hope that the revised manuscript meets your expectations.

Round 2
Reviewer 2 Report
Authors addressed points raised by reviewers.